# Immune Modulation Ability of Hepcidin from Teleost Fish

**DOI:** 10.3390/ani12121586

**Published:** 2022-06-20

**Authors:** Claudio Andrés Álvarez, Paula A. Santana, Nicolás Salinas-Parra, Dina Beltrán, Fanny Guzmán, Belinda Vega, Félix Acosta, Luis Mercado

**Affiliations:** 1Laboratorio de Fisiología y Genética Marina, Centro de Estudios Avanzados en Zonas Áridas, Coquimbo 1781421, Chile; belinda.vega@ceaza.cl; 2Facultad de Ciencias del Mar, Universidad Católica del Norte, Coquimbo 1781421, Chile; 3Instituto de Ciencias Químicas Aplicadas, Facultad de Ingeniería, Universidad Autónoma de Chile, San Miguel, Santiago 8910060, Chile; paula.santana@uautonoma.cl; 4Grupo de Marcadores Inmunológicos, Instituto de Biología, Pontificia Universidad Católica de Valparaíso, Valparaíso 2374631, Chile; nicolas.a.s.parra@gmail.com (N.S.-P.); dinabeltrancontreras@gmail.com (D.B.); 5Núcleo Biotecnología Curauma (NBC), Pontificia Universidad Católica de Valparaíso, Valparaíso 2374631, Chile; fanny.guzman@pucv.cl; 6Grupo de Investigación en Acuicultura (GIA), Instituto Universitario Ecoaqua, Universidad de Las Palmas de Gran Canaria, 35214 Telde, Islas Canarias, Spain

**Keywords:** hepcidin, host defense peptide, cytokines, *D. labrax*, *O. mykiss*

## Abstract

**Simple Summary:**

Antimicrobial peptides are part of the fish defense system, which can directly eliminate pathogenic microorganisms and, at the same time, regulate the immune response against them. This study evaluated the immunomodulatory effects of the antimicrobial peptide hepcidin in both juvenile fish and fish leukocyte cells. The results showed that hepcidin increased the expression of TNF-α, IL-1β, and IL-10 cytokines in leukocyte cells from trout. Moreover, the mRNA expressions of the same cytokines were up-regulated in different immune tissue of sea bass, confirming in vitro results. This study provides new insights into immunomodulatory function complementary to hepcidin’s previously established antimicrobial activity modulating the pro- and anti-inflammatory responses in teleost fish.

**Abstract:**

Antimicrobial peptides (AMP) play an essential role in the innate immune system, modulating the defense response. In a previous study, we demonstrated the antimicrobial activity of synthetic hepcidin (hep20) from rainbow trout (*Oncorhynchus mykiss*), and its protective effect in European sea bass (*Dicentrarchus labrax*) challenged with *Vibrio anguillarum*. Additionally, we described the uptake and distribution of hep20 in different tissues and leukocyte cells. Interestingly, various AMPs characterized in high vertebrates, called host defense peptides (HDPs), also possess immunomodulation activity. For that reason, the present study explores the immunomodulatory abilities of hep20 through in vitro and in vivo studies. First, a monocyte/macrophage RTS-11 cell line from rainbow trout was used to evaluate hep20 effects on pro- and anti-inflammatory cytokines in fish leukocyte cells. Next, the European sea bass juveniles were used to determine if hep20 can regulate the expression of cytokines in fish immune tissues. The results show that hep20 was uptake inner to RTS-11 cells and was able to induce the expression of IL-10, IL-1β, and TNFα at transcriptional and protein levels. Then, the European sea bass juveniles were given intraperitoneal injections of the peptide. At 1, 3, 7, 14, and 21 days post-injection (dpi), IL-10, IL -1β, and TNFα mRNA were quantified in the anterior gut, spleen, and head kidney. The hep20 was able to up-regulate cytokine gene expression in these tissues, mainly in the head kidney. Furthermore, the evaluated cytokines showed a cyclical tendency of higher to lesser expression. Finally, a bioinformatics analysis showed that the structure adopted by hep20 is similar to the γ-core domain described for cysteine-stabilized AMP, defined as immunomodulatory and antimicrobial, which could explain the ability of hep20 to regulate the cytokine expression. This study provides new insights into immunomodulatory function complementary to the previously established antimicrobial activity of hep20, suggesting a role as an HDP in teleost fish. These facts are likely to be associated with molecular functions underpinning the protective effect of fish hepcidin against pathogens.

## 1. Introduction

Antimicrobial peptides (AMPs) are small molecular weight compounds present in plants, insects, and vertebrates, forming part of their defense system. However, their name fails to describe all the functions they may exhibit. In higher vertebrates, it has been reported that they participate in other biological processes, such as tissue damage repair (scarring), regulation of metal absorption, immune response modulation, and even angiogenesis [1]. AMPs with these immunomodulatory functions, known as host defense peptides (HDPs), are expressed in different tissues and regulate various immune defense processes [2]. One of these peptides with interesting biological functions is hepcidin, a key molecule in iron absorption and homeostasis in different animals [3]. It is synthesized as a prepropeptide, mainly in the liver, and cytokines can induce its expression in response to pathogenic agents [4].

Studies on hepcidin in teleost fish have confirmed the relevance of the molecule in the defense system against pathogenic microorganisms [5,6]. In line with these studies, research in our laboratory has shown that two hepcidin variants from rainbow trout (*Oncorhynchus mykiss*) possess antimicrobial activity against important salmonid pathogens such as *Piscirickettsia salmonis*. Moreover, we observed that the variant with the amino-terminal amino acids glutamine-serine-histidine (QSH) (referred to as hep25) expressed the lowest antimicrobial activity, with Hep20 being the variant with the lesser number of amino acid residues in rainbow trout [7]. Nevertheless, two hepcidin variants from turbot (*Scophthalmus maximus*) showed that both peptides promoted antimicrobial and antiviral activity when the fish were infected with different microorganisms [8]. A similar result was obtained in a recent study on spotted scat (*Scatophagus argus*), describing the ability of hepcidin variants to inhibit the growth of pathogenic bacteria and protect against *Siniperca chuatsi* rhabdovirus and *Micropterus salmoides* reovirus [9].

Studies analyzing the expression of hepcidin in teleost fish show upregulation by the challengue with pathogen-associated molecular patterns (PAMPs), such as lipopolysaccharides and bacterial pathogens [7,10,11]. Moreover, it has been shown that they are up-regulated in the presence of the intracellular pathogen *Piscirickettsia salmonis* in the RTS-11 cell line from *O. mykiss* [10]. In addition, the activity of fish peptide variants with the QSH in the N-terminal may be associated with iron transporter interactions and iron absorption regulation [11], therefore inhibiting the growth of intracellular pathogens [3,7,12,13,14]. In this regard, we have previously found that this variant is induced in trout subjected to iron overload, suggesting that hep25 plays a role in iron homeostasis in a similar way than described in higher vertebrates [7,15].

As a result, the focus of the next studies was to analyze the antimicrobial effect of the Hep20 variant on rainbow trout. Firstly, in vitro activity assays were carried out, demonstrating the capacity of Hep20 to inhibit the growth of *P. salmonis* [7]. In that study, it was also shown that the peptide acts through a mechanism that is independent of the destabilization of the bacterial membrane, suggesting that it has an intracellular target [7]. In another study, the antimicrobial activity of Hep20 against *Vibrio anguillarum* was demonstrated, leading us to perform in vivo assays—challenging the sea bass with this pathogen. In that study, mortality rates were observed to be 50% lower among sea bass who received the Hep20 peptide [16]. In addition, it was shown for the first time that the absorption of this type of molecule occurs in teleost fish, since 30 min after intraperitoneal injection, the peptide was identified in the cytoplasm of gut cells. Interestingly, after 24 h, the peptide was also found in the spleen and the head kidney of the fish, which are specialized tissues in the immune response to pathogens [16]. These results suggest that the peptide is distributed systemically and that it may have an immunoregulatory effect on these tissues.

Considering the protective effect of Hep20 in sea bass challenged with *V. anguillarum*, the objective of the present study was to determine if the peptide can modulate the immune response in both fish immune tissues and fish leukocytes by influencing cytokine expression. Therefore, the focus was on evaluating the activity of the peptide in non-challenged fish/leucocytes to identify the effect of the molecule in the absence of indicators of a pathogen response, such as PAMPs or damage-associated molecular patterns (DAMPs). In addition, a bioinformatic analysis highlighted structural similarities between fish hepcidin and the multidimensional γ-core motif common to all classes of disulfide-stabilized AMPs, which could be associated with the immunomodulatory function of hepcidin in teleost fish.

## 2. Materials and Methods

### 2.1. Peptide Synthesis

Hep20 peptide was synthesized by a solid phase multiple peptide system using Fmoc L-amino acids (Iris and Rink resin 0.65 meq/g) and EMSURE^®^ organic solvent purchased from Merck Millipore (Merck KGaA, Darmstadt, Germany) [17]. The peptide was cleaved with TFA/TIS/EDT/H_2_0 (92.5/2.5/2.5/2.5) (trifluoroacetic acid/triisopropylsilane/1.2-ethanedithiol/ultrapure water, Merck KGaA, Darmstadt, Germany) and the purity was analyzed by RP-HPLC with a 0–70% acetonitrile-water mixture gradient over 30 min at a flow rate of 1 mL/min. The peptide was then lyophilized and analyzed using MALDI-TOF mass spectrometry (Bruker Daltonics, Billerica, MA, USA) to confirm its molecular mass, as previously reported [7]. The peptides were oxidized as previously reported [7]. Briefly, 5 mg of the crude peptide were dissolved in 50% (*v*/*v*) AcOH in H_2_O and subsequently diluted in 32 mL of oxidation buffer (10% isopropyl alcohol and 10% dimethyl sulfoxide). The pH of the peptide solution was adjusted to 5.8 with NH_4_OH and subjected to air oxidation at room temperature for 18 h; prior to Sep-Pak C18 purification, the pH was acidified to 2.5. The salts from the crude peptide were removed by elution in a Sephadex G-10 column (Waters Corporation, Milford, CT, USA). Finally, the peptides were applied onto a Sep-Pak C18 Vac cartridge (Waters Corporation, Milford, CT, USA) equilibrated in acidified water (0.05% trifluoroacetic acid in UPW-Ultra Pure Water). After washing with acidified water, the peptides were eluted at a flow-rate of 1 mL/min with 5%, 20%, 40%, 60%, and 80% acetonitrile (ACN) in water. The appropriate fractions were collected, and the ACN was evaporated in a SpeedVac centrifugal evaporator (Thermo Fisher Scientific, Waltham, MA, USA). The fractions were then analyzed using MALDI-TOF mass spectrometry.

### 2.2. Cell Culture and Incubation with Hep20

The RTS11 monocyte/macrophage cell line of *O. mykiss* (kindly donated by Dr. Niels Bols, University of Waterloo, Waterloo, ON, Canada) was cultured at 20 °C in Leibovitz’s L-15 medium (Gibco, Thermo Fisher Scientific, Waltham, MA, USA) supplemented with 15% fetal bovine serum (Gibco, Thermo Fisher Scientific, Waltham, MA, USA), as previously reported [18,19,20].

Cultured RTS-11 cells were treated with 10 μM Hep20 (suspended in phosphate-buffered saline (PBS)) for 3 and 6 h to determine if RTS11 cells uptake the synthetic peptide. After the incubation time, the cells were fixed on glass slides with 4% paraformaldehyde for 10 min and permeabilized with 0.5% Triton X-100 in PBS. Then, the cells were incubated with anti-hepcidin mouse antiserum (1:100) [10]. The samples were thoroughly washed with PBS and incubated with a 1:750 anti-mouse IgG Alexa-488 conjugated antibody (Invitrogen, Thermo Fisher Scientific, Waltham, MA, USA) for 1 h at room temperature. For membrane staining, samples were exposed to octadecyl rhodamine B chloride, R18 (Invitrogen, Thermo Fisher Scientific, Waltham, MA, USA) for 5 min at room temperature and then washed five times with PBS. Moreover, 4′,6-diamidino-2-phenylindole (DAPI) (Invitrogen, Thermo Fisher Scientific, Waltham, MA, USA) was used as nuclei stain.

To determine if Hep20 was able to induce cytokine mRNA expression in the RTS-11 cells, the cell cultures were incubated with 10 μM Hep20 (suspended in phosphate-buffered saline (PBS)) for 6, 12, and 24 h. Then, the cells were scraped from the flask and centrifuged at 300× *g* for 10 min. The cellular pellet was processed for RNA purification with the TRIzol LS Reagent (Invitrogen, Thermo Fisher Scientific, Waltham, MA, USA) and E.Z.N.A.^®^ Total RNA Kit II (Omega Bio-tek, Norcross, GA, USA) according to the manufacturer’s protocols. The RNA concentration was measured in a Nanodrop-1000 spectrophotometer (Thermo Fisher Scientific, Waltham, MA, USA). The mRNA expression levels of TNFα, IL1-β, and IL-10 were analyzed by RT-qPCR as described in Section 2.4.

To verify the effects of Hep20 on the cytokine expression at the protein level in the RTS-11 cells, incubation with 10 μM Hep20 for 24 h at 20 °C was performed. Then, the cytokines TNFα, IL1-β, and IL10 were detected by immunofluorescence as previously described, using mouse antisera against each of these molecules (1:100) and subsequent incubation with an anti-mouse IgG Alexa fluor 488 conjugated antibody (1:750). DAPI was used for nucleus staining.

All antisera used in this study have been previously validated for cytokine recognition in salmonid biological samples [21,22,23]. The images were obtained by epifluorescence microscopy (Leica) and then analyzed by the Leica Application Suite Advanced Fluorescence software (Leica, Wetzlar, Germany).

### 2.3. Experiment Fish and Hep20 Inoculation and Sample Collection

Eighty European sea bass with 110 g average body weight were acclimated in fiberglass tanks of 500 L at the Marine Science and Technology Park of Universidad de Las Palmas in Gran Canaria (Spain). All tanks were supplied with running seawater, constant aeration, and subjected to a photoperiod of 12 h of light and 12 h of darkness (12 h:12 h L:D). Fish were maintained at an average water temperature of 19.1 ± 0.3 °C, which was monitored daily twice daily.

After seven days of acclimatization, 36 fish were randomly distributed in three tanks per treatment (fish injected with 20 μg of Hep20 suspended in PBS and control fish injected with PBS). All tanks were supplied with running seawater, constant aeration, and natural photoperiod (12 h:12 h L:D). After 1, 3, 7, 14, and 21-days post-injection (dpi), six fish from each study group (two fish per tank) were sacrificed by an anesthetic overdose (200 mg L^−1^ of Tricaine MS-222 in seawater). Samples from the anterior gut, spleen, and head kidney were collected and stored with RNA and later frozen at −80 °C until gene expression analysis. The fish were not fed one day before being injected and one day before each sampling point.

### 2.4. RNA Extraction and RT-qPCR

RNA was extracted from European sea bass tissues using TRIzol LS Reagent (Invitrogen, Thermo Fisher Scientific, Waltham, MA, USA) and E.Z.N.A.^®^ Total RNA Kit II (Omega Bio-tek, Norcross, GA, USA) following the manufacturer’s instructions. RNA was quantified by nanodrop, and total RNA (1 μg) isolated from each sample was used as a template for cDNA synthesis (Script cDNA synthesis kit) (Bio-Rad Laboratories, Inc., Hercules, CA, USA). Quantitative RT-PCR (RT-qPCR) was performed with MyiQ Biorad using an SYBR Green Supermix (Bio-Rad Laboratories, Inc., Hercules, CA, USA). In the case of cytokine mRNA expression analysis of RTS-11 cells, the assays were carried out in an Mx3000P qPCR System (Agilent, Santa Clara, CA, USA). The sequences of the primers for investigating the cytokines expression in both fish species are shown in Appendix A. The results were expressed as relative amounts of the target gene using β-actin or EF-1α housekeeping gene to normalize the measured Cq values of target genes using the comparative Ct method (2^−ΔΔCt^) in the sea bass tissues and RTS11 cells, respectively.

### 2.5. In Silico Analysis of the Structure of Hep20

Multiple alignment of hepcidin from mammals and teleost fish was performed using software Clustal Omega (https://europepmc.org/article/MED/30976793, accessed on 7 May 2022) and analyzed with Jalview [24]. The sequence used for the alignment corresponded to *Mus musculus* 1 (Q9EQ21), *Mus musculus* 2 (Q80T19); *Homo sapiens* (P81172); *Rattus norvegicus* (Q99MH3); *Danio rerio* 1 (P61516), *Danio rerio* 2 (Q7T273); *Sus scrofa* (Q8MJ80); *Larimichthys_crocea* (A1Z0M0); *Morone chrysops* (P82951); *Oryzias latipes* (XP_004086682.1); *Oncorhynchus mykiss* (Q9DFD6) and *Dicentrarchus labrax* (A0A0G2SKR4). Using the free-online server SwissModel server (SWISS-MODEL (expasy.org), accessed on 7 May 2022), modeling of hepcidin from fish was performed with the human hepcidin as a template (PDB ID: 1M4F; 4QAE). The structure figures were prepared using Chimera v.1.16 (University of California, San Francisco, CA, USA) [25].

### 2.6. Statistical Analysis

Acclimation tanks were analyzed for a tank effect using a one-way analysis of variance (ANOVA). No differences were found between different experimental tanks, so individual fish were considered replicates in the analyses (*n* = 6).

The mRNA expression results are represented graphically using GraphPad prism 5 software (Dotmatics, San Diego, CA, USA) as mean ± standard deviation of the biological replicates as fold change compared to the control group in the first sampling point (day 1 post-injection).

One-way ANOVA followed by Tukey’s multiple comparison test was performed to compare the means of experimental groups (control and peptide) at different time points in sea bass tissues.

Results in the RTS11 cell line were expressed as fold change from control (untreated cells). Non-parametric Wilcoxon–Mann–Whitney test was used to compare the means of experimental groups.

All statistical analyses were performed using R version 3.6.1 (RStudio Inc, Boston, MA, USA). Differences were considered significant when *p* < 0.01 (**) or *p* < 0.05 (*).

## 3. Results

### 3.1. Sequence and Structure Analysis of Fish Hepcidin

Multiple alignments of teleost hepcidin with higher vertebrate’s hepcidin showed a highly conserved GXC motif, which is typical of peptides stabilized by disulfide bridges. Moreover, the structural signature called γ-core motif (NH2···C (X10-13) G (X3) C (X1-3) ···COOH) is present in this peptide, which appears to play a multifunctional role in the host defense described for cysteine-stabilized AMPs from different sources (Figure 1A).

The main characteristic of cysteine-rich peptides is their structure stabilization by forming disulfide bridges. In contrast, the glycine residue is involved in the β-hairpin loop to form the antiparallel β-strands, with both residues being highly conserved (Figure 1B,D). The folding of Hep20 includes the four disulfide bridges formed with the loss of eight hydrogen atoms resulting in an 8 Da decrease in molecular weight (predicted values: 2328.91 Da of reduced Hep20 to 2320.91 Da in the oxidation form). MALDI-TOF mass spectrometry detected the oxidized form of the synthetic Hep20 (Figure 1E).

In addition, an analysis of the surface hydrophobicity of Hep20 was performed. It shows a broad hydrophilic region (red), favoring its solubility in aqueous media and restricting the tendency for aggregation, which are major characteristics for determining its antimicrobial properties and probably the immune modulation (Figure 1C).

### 3.2. Cytokine Expression Profile in the Anterior Intestine of European Sea Bass Injected with Hep20

This study monitored the gene expression of cytokines for 21 days in sea bass given intraperitoneal injections of the synthetic peptide Hep20. Due to the way in which the peptide was administered, in the first instance, the expression of the cytokines IL-1β, IL-10, and TNFα were analyzed in the anterior intestine of the fish, observing a peak in TNFα expression at 3 dpi (Figure 2A). The transcript of this cytokine then decreased drastically at 14 and 21 dpi, reaching a level below that of the control group. The same kinetics of decrease were seen for the mRNA of IL-1β, though to a lesser extent at 7 dpi (Figure 2B), while the transcript of IL-10 showed a peak at 7 dpi and lower values at 14 and 21 dpi (Figure 2C).

### 3.3. Cytokine Expression Profile in the Spleen of European Sea Bass Injected with Hep20

The analysis of the spleen samples revealed similar expression kinetics for IL-1β and TNFα. However, a variable expression profile was observed in this tissue, with a significant increase in the mRNA content of both cytokines at 1 dpi and 14 dpi (Figure 3A,B). The transcript then decreased to values below those of the control group at 21 dpi (Figure 3A,B). In the case of IL-10, a significant increase was seen at 1, 7, and 14 dpi, while at 21 dpi, the expression of IL-10 decreased to values below those of the control group (Figure 3C).

### 3.4. Cytokine Expression Profile in the Head Kidney of European Sea Bass Injected with Hep20

The highest level of expression for the mRNA of these cytokines was obtained in the head kidney of sea bass. An expression peak was seen for the mRNA of IL-1β and TNFα at 3 dpi, followed by a gradual decline in expression at 7 and 14 dpi and a more drastic decrease at 21 dpi (Figure 4). In the case of IL-10, a peak in the expression of its mRNA was seen at 7 dpi, while the lowest expression levels were found at 21 dpi (Figure 4C).

### 3.5. Cytokine Expression Induced by Hep20 in Cell Culture

The effects of Hep20 on the RTS11 cell line were first assessed by determining whether the synthetic peptide was able to enter the cell. Images of immunofluorescence indicate the Hep20 location on the cytosol and perinuclear area of RTS11 cells. In addition, after 6 h post-treatment, the detection of Hep20 was increased in RTS11 cells (Figure 5A).

The ability of Hep20 to induce cytokine expression in the RTS11 cell line was evaluated by RT-qPCR and immunodetection of IL-10, IL-1β, and TNFα. By RT-qPCR, the mRNA expression of these cytokines was quantified after 6, 12, and 24 post-treatment with Hep20. The relative expression indicates that the highest up-regulation occurred after 24 post-treatment with the synthetic peptide for all cytokines tested (Figure 5B). Moreover, at an early time (6 h), the expression of the pro-inflammatory cytokine TNF-a is significantly up-regulated. At 12 h, there was a non-significant upregulation trend in the expression of all cytokines tested. For that reason, the protein detection of IL-10, IL-1β, and TNFα was performed 24 h post-treatment with Hep20. The immunofluorescence image shows that only the pro-inflammatory cytokines IL-1β and TNFα can be detected in the cytosol of RTS11 cells (Figure 5C). Although the mRNA expression of IL-10 anti-inflammatory cytokine was induced at 24 h, it was not detected at the protein level.

## 4. Discussion

Traditionally, cationic peptides have been studied as antimicrobial agents. However, their direct on microbial death may not be their unique function. This is firstly because the physiological concentrations of these peptides differ in magnitude from the concentrations used in the analysis of their antimicrobial activity in vitro, and because monovalent and bivalent cations, serological factors, and polyanionic saccharides, which are often antagonists to their activity, are absent or present at lower concentrations in such studies [26,27]. Thus, the conditions under which these assays were carried out do not accurately reflect the tissue environment in which the cationic peptides perform their antimicrobial functions. All the above led us to posit that their primary action may be more related to the modulation of the immune response [26,28]. In line with these studies, we have observed that the peptide Hep20 derived from trout hepcidin is able to increase the gene expression of some cytokines in the teleost fish model. In general, it was seen that the expression of the pro-inflammatory cytokines IL-1β and TNFα was induced at 3 dpi in the gut of the juvenile sea bass, while in more specialized immune response tissues, such as the head kidney and spleen, the increase was seen from 1 dpi. Furthermore, the highest expression values for these cytokines were observed in the head kidney, with maximum induction at 3 dpi.

The results obtained in this study compare well with experiments with sea bass performed previously in our laboratory, challenging the fish with *V. anguillarum* [16]. In that test, the fish began to die from vibriosis at 3 dpi. However, a 50% decrease in mortality was observed for those that were injected with the peptide. Hence, the results obtained in the present study suggest that Hep20 may protect from vibriosis by stimulating inflammatory processes, thus contributing to host defense mechanisms to avoid colonization and invasion of *V. anguillarum* into the fish’s tissues.

TNFα induction is one of the key features of resistance to *V. anguillarum* in the inflammatory processes activated by Hep20. It has been shown that the application of rTNFα to sea bass significantly increased the protection afforded by the oral vaccines against vibriosis [29]. According to that study, TNFα activates the CCL25/CCR9 ligand/receptor system in the gut, resulting in an increase in the differentiation and migration of T-cytotoxic cells, strengthening the action of vaccines that protect against *V. anguillarum* [29]. Nevertheless, it remains to be studied if Hep20 has a similar effect on the action of oral vaccines.

Along with the increase in IL-1β and TNFα, there was also an increase in the expression of the transcript of IL-10. In this study, the head kidney is the tissue with the highest increase in inflammatory cytokine expression. Nevertheless, the peak expression of IL-10 was seen at 7 dpi, 4 days after the highest levels of expression for IL-1β and TNFα occurred. This overexpression may be related to the regulation of the inflammatory response, ensuring that the process does not take place excessively [30,31]. Perhaps, the anti-inflammatory response observed in the head kidney could be a consequence of the upregulation of pro-inflammatory cytokines triggered by Hep20; therefore, an anti-inflammatory response is caused by the fish itself.

Several studies suggest that the cationicity of AMPs is fundamental to the initial attraction to the electronegative membranes of the pathogens [32]. This characteristic has aroused interest in biotechnology companies regarding the use of AMPs as potential therapeutic agents [33]. However, despite the high number of different AMPs identified, only a few of them have been studied in their capacity to regulate host innate immunity [32]. Evidence suggests that the immunomodulatory properties of some cationic peptides are fundamental for coordinating the defense mechanisms of different organisms. These molecules are able to regulate the immune response by increasing or decreasing cytokine expression and, at the same time, inhibiting the growth of microorganisms [34,35]. This is the case of epicidin-1, an AMP isolated from a marine grouper, which has antimicrobial activity against Gram-negative and Gram-positive bacteria, and was later described as an immunomodulatory peptide capable of increasing the plasmatic concentration of IL-10 in mice with expression peaks at 1, 2, and 16 dpi [36]. Hence, epicidin-1 can attenuate the inflammatory response in mice, protecting against LPS-induced organ damage [36]. This peptide has also shown immunomodulatory properties in zebrafish, conferring it protection against infection with *Vibrio vulnificus.*

To date, the mechanism by which different AMPs modulate immune response remains unknown [37]. There is evidence that they can even inhibit the activation and differentiation of dendritic cells [38,39,40]. It has been posited that due to their amphipathic properties, they may pass through the bacterial membrane and alter signaling pathways, such as the phosphorylation of MEK1/2, or they may alter secondary messenger routes, such as cAMP, thus modulating the activation of transcription factors, one of which is NF-kB [41]. Considering the ability of some AMPs to penetrate cell membranes, Hep20 was located after RTS-11 cell line treatment with the synthetic peptide. IFAT showed that Hep20 was clearly located on RTS11 cell cytoplasm at 6 h post-treatment. Therefore, future studies will be aimed to elucidate if Hep20 can regulate fish leukocyte signaling pathways. Consequently, IL-1β and TNFα expression was significantly increased in cells treated with Hep20; this result indicates that Hep20 may regulate the leukocyte cytokine response to up-regulate pro-inflammatory response, which permits enhanced antimicrobial capacity of fish immune cells. In addition, the leukocyte in vitro results is in concordance with the in vivo experiment because a robust pro-inflammatory response was also obtained in the head kidney and spleen, the primary leukocyte tissues of teleost fish.

In recent years, new evidence has been gathered recognizing cysteine-rich AMPs as potent chemoattractant-inducers; for example, β-defensins attract cells by interaction mainly with CCR2 and CCR6 receptors [42]. Surprisingly, the 3D structures of trout and sea bass hepcidin are essentially superimposable over the γ-core domain, which is described for disulfide-stabilized AMPs. Current findings suggest that iterations of the γ-core motif are common to broader classes of disulfide-stabilized host defense effector molecules, not only in AMPs, but also in peptide toxins, plant thionins, and chemokynes [43,44,45]. Thus, the preservation of the γ-core motif structure, despite a high level of sequence variability, may have enabled the evolution of a broad range of HDPs with additional and/or specialized activities [43,44,45]. Therefore, the understanding of hepcidin as an HDP could be connected to the type of structure it adopts, which requires further analysis of immunomodulatory properties, such as chemotaxis or the direct interaction with cytokine receptors.

## 5. Conclusions

This study has found evidence that teleost hepcidin, together with its antimicrobial activity, may have the potential to generate more robust immunological responses by the regulation of mRNA expression of both pro- and anti-inflammatory cytokines. Future studies will be focused on further exploration of such peptides as possible adjuvants in the control of infectious diseases that constantly affect fish farming. In addition, new assays will be required to explore the potential action mechanism associated with its immunomodulatory properties.

## Figures and Tables

**Figure 1 animals-12-01586-f001:**
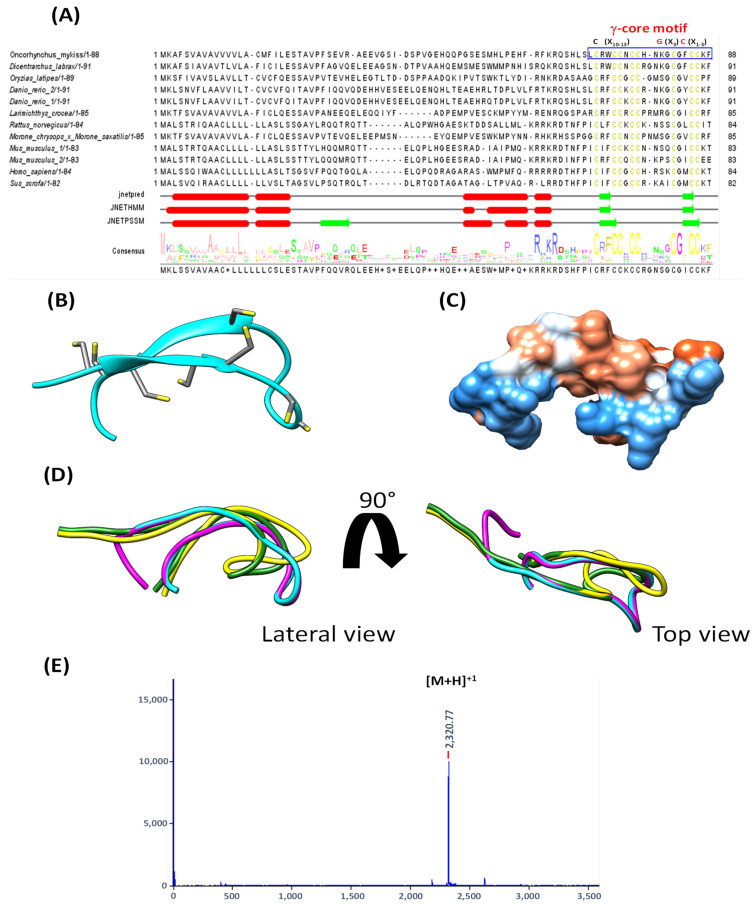
Hepcidin sequence analysis and synthesis of Hep20. (**A**) Multiple alignment analysis of hepcidin from fish and vertebrates. The blue box highlights the sequence of synthetic peptides (Hep20) from *Oncorhynchus mykiss*. Cysteine residues are shown in yellow. The GXC motif is shown in red, which is conserved in the different species being part of the γ-core motif. (**B**) A secondary structure prediction indicates that these peptides are composed of β-strand at the C-terminus. Cysteine residues of Hep20 are shown as stick representation (yellow). (**C**) Representation of hydrophobic (blue) and hydrophilic (red) surfaces of Hep20. (**D**) Superposition of C-terminus segment of *Oncorhynchus mykiss* (cyan), *Oryzias latipes* (green), *Dicentrarchus labrax* (yellow), and *Homo sapiens* (pink, 1M4F). (**E**) MALDI-TOF mass spectrum obtained for the synthetic Hep20 peptide, which confirms its molecular mass in an oxidized state showing the charged ion [M+H]^+^ (*m*/*z* 2320.77).

**Figure 2 animals-12-01586-f002:**
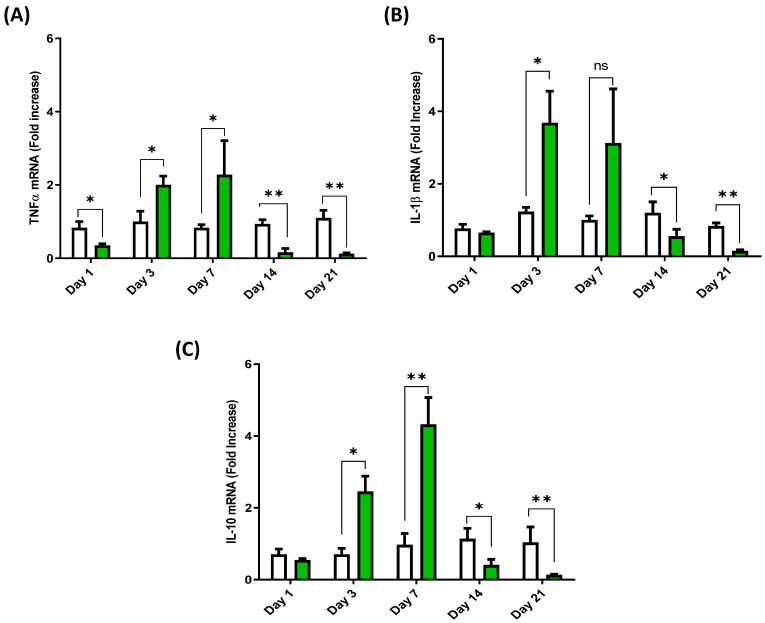
Anterior intestine cytokine mRNA expression profile of European sea bass injected with Hep20. TNFα (**A**), IL-1β (**B**) and IL-10 (**C**) mRNA expression in the anterior intestine of European sea bass after Hep20 intraperitoneal injection (green bars). The graph shows the cytokine fold induction compared to the control group, i.e., fish injected with phosphate-buffered saline (white bars). Significant differences (*p* < 0.05 or *p* < 0.01) between groups at each time are indicated by (*) or (**) respectively (*n* = 6), ns (non-significant differences).

**Figure 3 animals-12-01586-f003:**
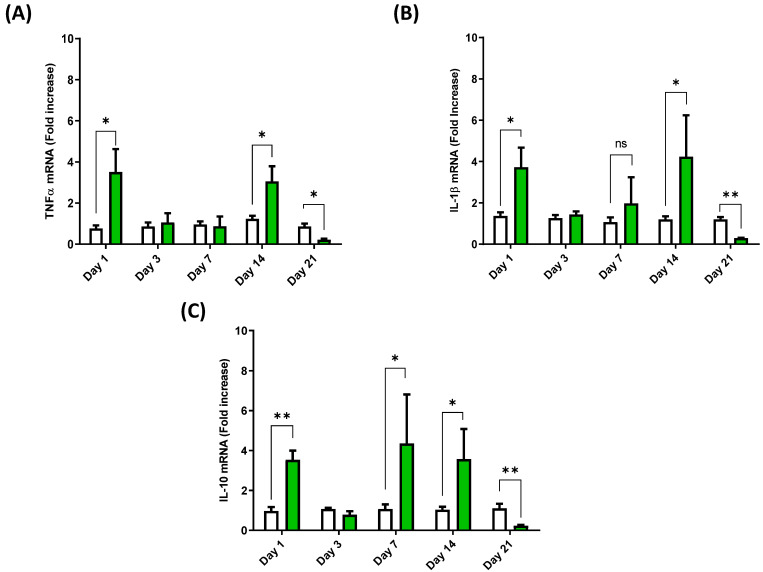
Spleen cytokine mRNA expression profile of European sea bass injected with Hep20. TNFα (**A**), IL-1β (**B**) and IL-10 (**C**) mRNA expression in the spleen of European sea bass after Hep20 intraperitoneal injection (green bars). The graph shows the cytokine fold induction compared to the control group, i.e., fish injected with phosphate-buffered saline (white bars). Significant differences (*p* < 0.05 or *p* < 0.01) between groups at each time are indicated by (*) or (**) respectively (*n* = 6), ns (non-significant differences).

**Figure 4 animals-12-01586-f004:**
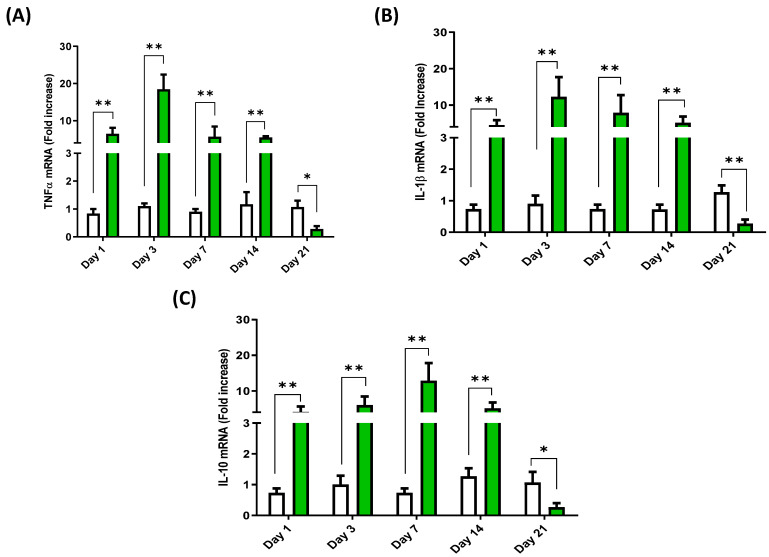
Head kidney cytokine mRNA expression profile of European sea bass injected with Hep20. TNFα (**A**), IL-1β (**B**) and IL-10 (**C**) mRNA expression in the head kidney of European sea bass after Hep20 intraperitoneal injection (green bars). The graph shows the cytokine fold induction compared to the control group, i.e., fish injected with phosphate-buffered saline (white bars). Significant differences (*p* < 0.05 or *p* < 0.01) between groups at each time are indicated by (*) or (**) respectively (*n* = 6).

**Figure 5 animals-12-01586-f005:**
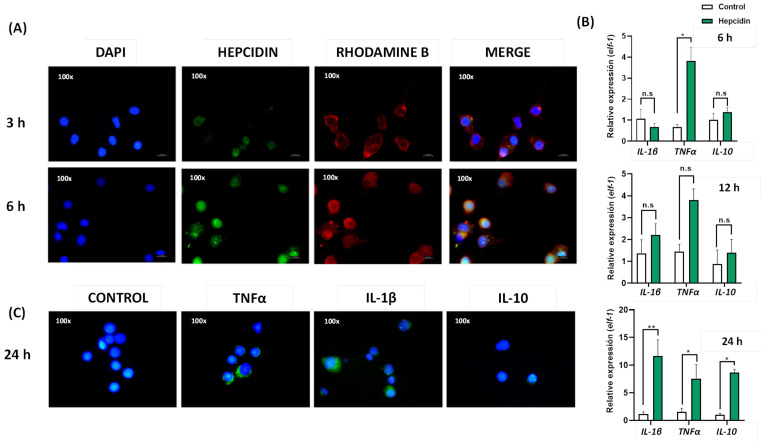
Effect of Hep20 on the expression of IL-10, IL-1β, and TNFα cytokines in RTS-11 cell line. (**A**) RTS11 cells treated with Hep20. Hepcidin was detected using mouse-specific antiserum and the anti-mouse IgG Alexa-488 conjugated antibody (green). Octadecyl rhodamine B chloride (R18) was used for membrane staining (red). Moreover, DAPI was used as nuclei staining (blue). (**B**) Relative mRNA expression of IL-1β, TNFα, and IL-10 mRNA in RTS-11 cell line treated with Hep20 for 6, 12, and 24 h. The mRNA expression is expressed as a fold-change over the control (non-treated cells). Data are presented as the mean ± standard error, with *n* = 4. * *p* < 0.05, ** *p* < 0.01 from Wilcoxon–Mann–Whitney non-parametric test; n.s (non-significant differences). (**C**) Detection of IL-1β, TNFα, and IL-10 on RTS11 cells after 24 h post-treatment with Hep20. The cytokines were detected using mouse-specific antiserum and the anti-mouse IgG Alexa-488 conjugated antibody (green). Moreover, DAPI was used as nuclei staining (blue).

## Data Availability

The data presented in this study are available on request from the corresponding authors.

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
