# Peer review of "Immune Modulation Ability of Hepcidin from Teleost Fish"

_animals, 2022, doi:10.3390/ani12121586_

Round 1

Reviewer 1 Report

This article written for Animals journal is an investigation into the immunomodulatory role of hepcidin-20 in teleost fish. The data presented in support of the immunomodulatory role of hepcidin-20 is clear, and includes consideration of several time points, tissue types and methodologies (invivo, invitro and insilico). This report will be valuable to fish biologists, immunologists and aquaculturists. This reviewer has a few clarification points/comments, as noted below:

1.     Hepcidin 20 used in the studies was synthetically synthesized. Did the authors conduct any validation assays to ensure that this was indeed a biologically active compound? Could this compound be availed naturally and used as a positive control?

2.     How did the investigators ensure that the Hep-20 preparation used in these studies was LPS free? Was the control (PBS) the same as the vehicle in which Hep-20 was dissolved?

3.     What was the rationale for selecting Hep-20 dosages for in-vitro and in-vivo analysis? Did the investigators test other dosages, besides the one listed?

4. The authors are encouraged to note the supplier of reagents (e.g. cell lines) noted in the methods. 

Author Response

This article written for Animals journal is an investigation into the immunomodulatory role of hepcidin-20 in teleost fish. The data presented in support of the immunomodulatory role of hepcidin-20 is clear, and includes consideration of several time points, tissue types and methodologies (invivoinvitro and insilico). This report will be valuable to fish biologists, immunologists and aquaculturists. This reviewer has a few clarification points/comments, as noted below:

  1. Hepcidin 20 used in the studies was synthetically synthesized. Did the authors conduct any validation assays to ensure that this was indeed a biologically active compound? Could this compound be availed naturally and used as a positive control?

Response: The biological activity of hepcidin is associated with its folding. The methodology section describes the oxidization procedure after being synthesized for the disulfide bridges that keep their conformation stable are formed. This protocol was validated in previous works where the beta-sheet structure after oxidation was visualized by circular dichroism (1). In addition, the antimicrobial activity of the synthetic peptide was also validated in previous studies, both in vitro (bacterial cultures) and in vivo (a challenge with V. anguillarum) (1,2).

Unfortunately, extracting hepcidin from the fish's biological fluids is complicated since, in previous studies, it was only possible to identify the molecule in trout tissues and blood through highly sensible techniques such as ELISA Sandwich or mass spectrometry (3,4). Therefore, it would be necessary to sacrifice several fish to obtain an adequate quantity to be used in bioassays exposed in this work.

  1. Alvarez, C.A.; Guzmán, F.; Cárdenas, C.; Marshall, S.H.; Mercado, L. Antimicrobial Activity of Trout Hepcidin. Fish Shellfish Immunol. 2014, 41, 93–101, doi:10.1016/j.fsi.2014.04.013.
  2. Álvarez, C.A.; Acosta, F.; Montero, D.; Guzmán, F.; Torres, E.; Vega, B.; Mercado, L. Synthetic Hepcidin from Fish: Uptake and Protection Against Vibrio anguillarum in Sea Bass (Dicentrarchus labrax). Fish Shellfish Immunol. 2016, 55, 662–670, doi:10.1016/j.fsi.2016.06.035.
  3. Santana, P.A.; Álvarez, C.A.; Guzmán, F.; Mercado, L. Development of a Sandwich ELISA for Quantifying Hepcidin in Rainbow Trout. Fish Shellfish Immunol. 2013, 35, 748–755, doi:10.1016/j.fsi.2013.06.005.
  4. Álvarez, C. a; Santana, P.A.; Guzmán, F.; Marshall, S.; Mercado, L.A. Detection of the Hepcidin Prepropeptide and Mature Peptide in Liver of Rainbow Trout. Dev. Comp. Immunol. 2013, 41, 77–81, doi:10.1016/j.dci.2013.04.002.
  5. How did the investigators ensure that the Hep-20 preparation used in these studies was LPS free? Was the control (PBS) the same as the vehicle in which Hep-20 was dissolved?

Response: Our peptide synthesis uses solvents of the highest degree of purity, ensuring its production is free of toxins. Therefore, we have added a sentence regarding the solvents used in the synthesis (lines 117-118). We also clarify that the peptide was suspended in PBS before being applied to fish and cell lines (line 141 and 174).

Additionally, we believe it is important to mention that LPS contamination occurs when the molecules when the molecules are produced by recombinant proteins in bacterial models. Hep20 was obtained by chemical synthesis, and the purity of the peptide was confirmed by standard HPLC analysis (1,3,4).

  1. What was the rationale for selecting Hep-20 dosages for in-vitro and in-vivo analysis? Did the investigators test other dosages, besides the one listed?

Response: Our previous work allowed us to choose the appropriate dose to observe biological effects in vivo and in vitro (1,2). Regarding studies in fish, our previous work with similar doses in sea bass managed not only to verify the ability to protect fish from vibriosis but also to be able to detect the peptide applied exogenously in immune response tissues such as the anterior kidney and spleen (2). However, it is also important to consider that the costs of the synthesis and oxidation of the molecule are expensive, limiting the number of fish we can inject at the indicated dose and testing other doses.

  1. The authors are encouraged to note the supplier of reagents (e.g. cell lines) noted in the methods.

Response: Sorry for this mistake. All reagents supplier were added.

Reviewer 2 Report

Dear Authors,

I found your study quite interesting even if a very research question is not well exposed in the introduction section. There are several minor revisions dispersed in the text but the study is affected by a big major problem related to experimental design, as shown by paragraph 2.3, evidently the study did not include replicas. It's therefore not possible to consider it for publication, I'm sorry but I hope that you comprise and understand.

Best regards

The Reviewer

Author Response

I found your study quite interesting even if a very research question is not well exposed in the introduction section. There are several minor revisions dispersed in the text but the study is affected by a big major problem related to experimental design, as shown by paragraph 2.3, evidently the study did not include replicas. It's therefore not possible to consider it for publication, I'm sorry but I hope that you comprise and understand.

Response: We have better exposed our experimental design in the new version of the manuscript. We worked with three culture tanks per treatment (fish injected with Hep20 suspended in PBS and control fish injected with PBS). Two individuals were removed from each pond at each sampling time. The number of fish was adjusted by the availability of the bioactive peptide form described in the first answer for Reviewer 1.

In addition, acclimation tanks were analyzed for a tank effect using a one-way analysis of vari-ance (ANOVA). No differences were found between different experimental tanks, so individual fish were considered as replicates in the analyses (n=6).       

Thus, we believe that our experimental design supports the results discussed in the article.

Author Response

The manuscript entitled “Immune modulation ability of hepcidin from teleost fish” by Claudio Andrés Álvarez et al. describing an attempt to provide a valuable insight. Antimicrobial peprides can regulate the pro-and anti-inflammatory responses in teleost fish. The manuscript is well written, however the objectives of the study were not clear in some section.

  1. RTS11 cell is used in in vitro experiments, but why is European sea bass used in in vivo

experiments?

Response: For the in vitro study (RTS-11 cell line, ref 1,2) and in vivo (European sea bass, 3), there are previous studies by the working group that supports its use to evaluate the activity of the Hep20 peptide.

The work was carried out simultaneously by laboratories in Chile and Spain, for which the results complement each other to expose the ability of this peptide as an immunomodulator in teleost fish. In our subsequent works, we hope to deepen the activity of this molecule as a potential fish vaccine adjuvant, where it is expected to add salmonids species and compare with the results of European sea bass.

  1. Álvarez, C.A.; Gomez, F.A.; Mercado, L.; Ramírez, R.; Marshall, S.H. Piscirickettsia salmonis Imbalances the Innate Immune Response to Succeed in a Productive Infection in a Salmonid Cell Line Model. PLoS One 2016, 11, e0163943, doi:10.1371/journal.pone.0163943.
  2. Alvarez, C.A.; Guzmán, F.; Cárdenas, C.; Marshall, S.H.; Mercado, L. Antimicrobial Activity of Trout Hepcidin. Fish Shellfish Immunol. 2014, 41, 93–101, doi:10.1016/j.fsi.2014.04.013.
  3. Álvarez, C.A.; Acosta, F.; Montero, D.; Guzmán, F.; Torres, E.; Vega, B.; Mercado, L. Synthetic Hepcidin from Fish: Uptake and Protection Against Vibrio anguillarum in Sea Bass (Dicentrarchus labrax). Fish Shellfish Immunol. 2016, 55, 662–670, doi:10.1016/j.fsi.2016.06.035.

  1. Line 167 to 176, What is the temperature of the breeding water for the European sea bass and

whether they were fed during the experiment period ?

Response:  Fish were maintained at 19.1 ± 0.3 °C and were not fed one day before being injected and one day before each sampling point. This information was added in the new version of the manuscript.

  1. Line 174 to 176, Whether the fish were anesthetized before sacrificed ?

Response:  Fish were immediately sacrificed by an anesthetic overdose (200 mg Lˉ 1 of Tricaine MS-222 in seawater). This information was added in the new version of the manuscript.

  1. In figure 5. (B), the middle panel should be used for statistical analysis.

Response: Sorry for this mistake. It was corrected in the new version of the manuscript.

  1. Why is there no consistency in the time points of figure 5. (A), (B), (C) observation and analysis ?

Response: The treatment times in Figure 5 correspond to the assessment of different processes.

In the case of Figure 5A, the ability of hepcidin to enter the cell is shown, allowing us to verify that its possible immunomodulatory effects could be due to intracellular targets. The fluorescence images were taken at a shorter time (3 and 6 hours post hepcidin application) because we needed to observe where hepcidin is after being applied in RTS11 cells. The images show the location of hepcidin inside RTS11 cells at both times.

In the case of Figure 5B, the results show the effect of hepcidin on relative gene expression of cytokines; therefore, the immunomodulatory role of hepcidin in RTS11 cells. In this case, because we need to analyze the transcriptional machinery activation of RTS11 cells after hepcidin treatment, the 6 hours post peptide application was considered as a first sample time. Then, the relative expression of immune genes was measured at 12 and 24 hours to verify its effects over time.

In the case of Figure 5C, the 24-hour time point was chosen because the gene expression results in Figure 5B showed significant increasing gene expression of the three target cytokines, so we assessed whether this could be observed at the protein level.

In brief,  the time points in A, B, and C are different because the objectives were different. Firstly, the analysis of Hep20 location on RTS11 cells, then evaluate the immunomodulatory effects at the transcriptional level, and finally verify the immunomodulatory effects at the protein level, respectively.

Round 2

Reviewer 2 Report

Dear Authors, 

thanks to have clarified my doubts related to the experimental desing, that now appears understandable. The first version of manuscript was lacking of a lot of important information, especially in the materials and methods section, but looking also on the other reviewer reports and your replies, I saw that some other my possible comments were altready solved.

Best regards

The Reviewer